# Examination of the proximodistal patellar position in small dogs in relation to anatomical features of the distal femur and medial patellar luxation

**Sawako Murakami** * , **Masakazu Shimada, Yasuji Harada, Yasushi Hara**

Department of Veterinary Surgery, Division of Veterinary Science, Section of Veterinary Medicine, Nippon Veterinary and Life Science University, Musashino, Tokyo, Japan

* muracarmin@gmail.com

**Data Availability Statement:** All relevant data are within the manuscript and its Supporting Information files.

## Abstract

### Objective

To determine the influence of anatomical features of the distal femur on the proximodistal patellar position and compare the proximodistal patellar position between dogs with and without medial patellar luxation (MPL).

### Study design

Retrospective case series (n = 71).

### Methods

Mediolateral-view radiographs of clinical cases of dogs weighing less than 15 kg were obtained. The stifle joint angle, patellar ligament length, patellar length, size of the femoral condyle, trochlear length, and trochlear angle were measured and included in multiple linear regression analyses to ascertain their effects on the proximodistal patellar position. Radiographs were divided into MPL and control groups. The effects of MPL on the proximodistal patellar position and morphological factors were also examined.

### Results

The final model for the proximodistal patellar position revealed that the patella became distal as the ratio of the patellar ligament length to patellar length decreased, the trochlear angle relative to the femur increased, the trochlear length relative to the patellar length increased, or the trochlear length relative to the femoral condyle width decreased. The proximodistal patellar position in the MPL group was not significantly different from that in the control group despite the trend towards a distally positioned patella (p = 0.073). The MPL group showed a significantly shorter trochlea (p<0.001) and greater trochlear angle relative to the femur (p = 0.029) than the control group.

**Funding:** The authors received no specific funding for this work.

**Competing interests:** The authors have declared that no competing interests exist.

## Conclusion

The proximodistal patellar position depends on multiple factors, and its determination based on PLL/PL alone may not be appropriate. Dogs with MPL did not have a proximally positioned patella compared with dogs without MPL. Although hindlimbs with MPL had a shorter trochlea than those without patellar luxation, this difference did not appear to be sufficient to displace the patellar position proximally in small dogs, possibly compensated by increased trochlear angle relative to the femur.

## Introduction

The proximodistal patellar position has long been discussed in relation to patellar luxation in dogs [1–8], although it is difficult to assess since it changes with stifle flexion or extension [8]. Thus, the ratio of the patellar ligament length to patellar length (PLL/PL) based on the Insall-Salvati index in humans was introduced as an index that is minimally affected by the stifle angle [6, 9–11]. Although some studies have introduced other indices to evaluate the proximodistal patellar position [2, 12, 13], these indices have not been used in clinical cases as often as the PLL/PL. Many canine studies have focused on the PLL/PL to discuss the proximodistal alignment between the patella and the femoral trochlea [1–8]. As the PLL increases, the patellar position becomes more proximal. However, other morphological factors, such as the size of the femoral condyle or the shape of the trochlea, can affect the proximodistal patellar position. One study in humans evaluated the association between the Insall-Salvati index and the cartilage congruence of patellofemoral joint measured on MRI and showed only a weak correlation [10]. Because the morphological features of dogs are more diverse than those of humans, the effect of the PLL/PL on the proximodistal patellar position in dogs could be smaller, and other morphological features might have an effect. One previous study suggested that the long proximal tibia might be the reason for the distally positioned patella of large dogs with lateral patellar luxation instead of a short patellar ligament [5]. However, no study has evaluated the effect of the anatomical features of the distal femur on the proximodistal patellar position in dogs.

In humans, a proximally positioned patella (patella alta) is associated with patellar instability, due to which the patella exceeds the femoral trochlea proximally and loses support from the trochlear ridge [11, 14–16]. Based on these human studies, veterinary researchers have investigated the association between the PLL/PL and patellar luxation [2–7]. Some studies found that large dogs with medial patellar luxation (MPL) had a larger PLL/PL than dogs without patellar luxation [5, 7]. However, other studies found no such difference between small-breed dogs with and without MPL [4, 17]. If a proximodistal patellar position is influenced by factors other than the PLL/PL, it would be prudent to examine the proximodistal patellar position itself between dogs with and without patellar luxation.

Thus, the first objective of this study is to determine the influence of the anatomical features of the distal femur on the proximodistal patellar position relative to the trochlea, and the second objective is to compare the proximodistal patellar position between small dogs with and without MPL. The first hypothesis is that the anatomical features of the distal femur, as well as the PLL/PL, will affect the proximodistal patellar position. The second hypothesis is that the proximodistal patellar position relative to the trochlea is more proximal in dogs with MPL, if the first hypothesis is validated.

## Materials and methods

Radiographs of privately owned dogs presented to the Nippon Veterinary and Life Science University or Minagawa Pet Clinic were evaluated retrospectively. Informed consent was obtained from all owners. This study was exempted from review by our ethics committee, according to the guidelines of the Nippon Veterinary and Life Science University Medical Ethics Review Committee. Radiographs of hindlimbs with orthopaedic disease other than MPL, or with a history of any stifle surgery were excluded from the study. Breed, sex, age, body weight, limb side, and grade of MPL were obtained from the patient records. Mediolateral-view radiographs taken with an exact overlap of the lateral and medial condyles were included in the study. Radiographs in which the patella was not in the trochlea were excluded; thus, no dog with grade 4 patellar luxation was included. Radiographs of hindlimbs with MPL were classified into the MPL group and radiographs of hindlimbs without any orthopaedic disease were classified into the control group.

### Measurements

Based on a previously reported method with few modifications for the application to small-breed dogs, the stifle joint angle was defined as the caudal angle made by the anatomical axes of the distal femur and the proximal tibia (Fig 1) [5]. The distal femoral anatomical axis was defined as the extension of the line connecting the two centres of the femoral width. The distal femoral width was one femoral-condyle length away from the proximal extremity of the trochlea, and the proximal width was half the length of the femoral condyle away from the distal one. The proximal tibial anatomical axis was defined as the extension of the line connecting the centre of the tibial width, which was 1.5 times the length of the proximal tibial width away from the tibial plateau, and the notch in front of the tibial plateau. The PL was measured as the

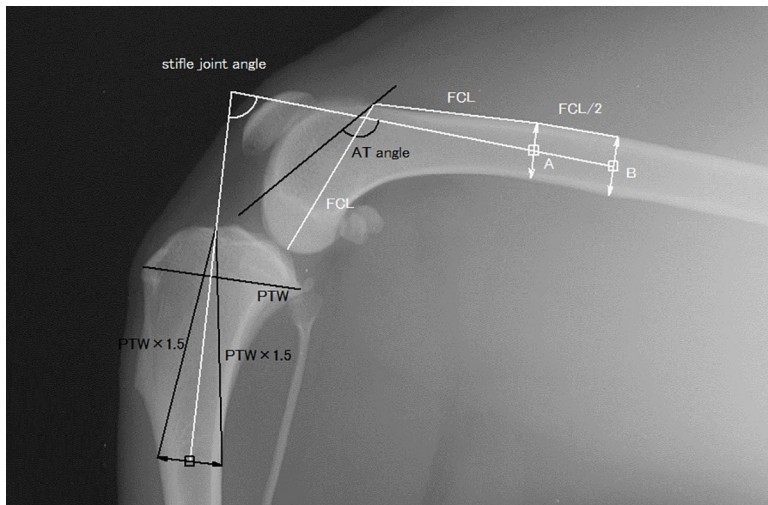

**Fig 1. The measurement definitions of the joint angle and anatomical trochlear angle (AT angle).** The joint angle (white arc) was measured as the caudal angle of the anatomical axes of the femur and tibia. The distal femoral anatomical axis was defined as the extension of the line connecting the two points (white squares, A and B). The distal point A was the centre of the femoral width, which was one femoral condyle length (line FCL) away from the proximal extremity of the trochlea, and the proximal point B was half the FCL away from the distal point. The proximal tibial anatomical axis was defined as the extension of the line connecting the centre of the tibial width (black square) and the notch at the front of the tibial plateau. The tibial width was measured at the tibial cortex, which was 1.5 times the length of the proximal tibial width (PTW) away from the notch at the front of the tibial plateau. The AT angle was measured as the caudal angle made by the femoral anatomical axis and the femoral trochlear line.

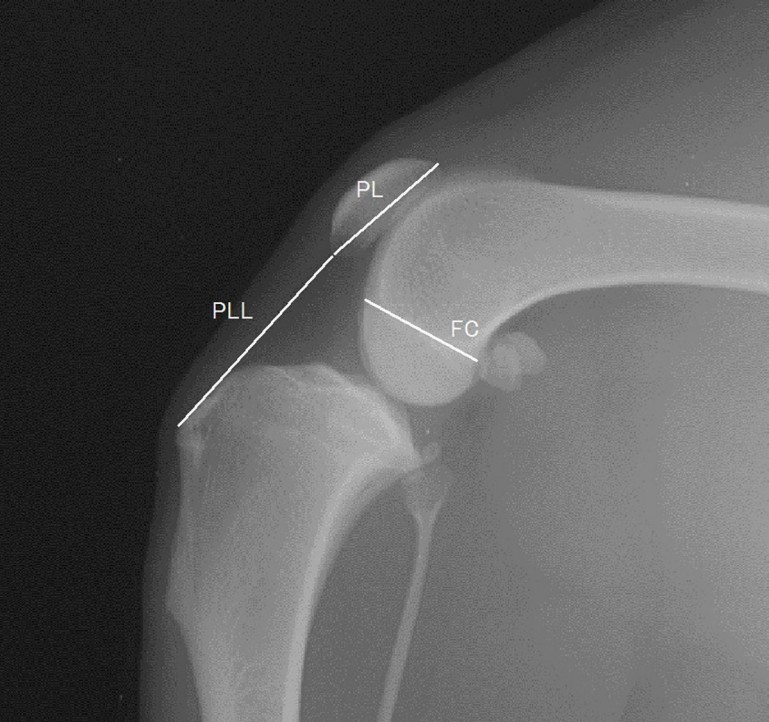

**Fig 2. The measurement definitions of the patellar length (PL), patellar ligament length (PLL), and size of the femoral condyle (FC).** The PL was measured as the longest dimension of the patella. The PLL was measured from the most distal portion of the patella to the patellar ligament insertion on the cranioproximal portion of the tibial tuberosity. The FC was measured from the origin of the long digital extensor muscle to the caudal femoral cortex along the Blumensaat's line.

longest dimension of the patella. The PLL was measured from the most distal portion of the patella to the patellar ligament insertion on the cranioproximal portion of the tibial tuberosity (Fig 2) [5]. The craniocaudal size of the femoral condyle (FC) was measured from the origin of the long digital extensor muscle to the caudal femoral cortex along the Blumensaat's line (Fig 2). The femoral trochlear length (TL) was measured from the proximal extent of the femoral trochlear ridges to the origin of the long digital extensor muscle, as in previous studies (Fig 3) [5, 8]. The anatomical trochlear angle (AT angle) was defined as the caudal angle between the distal femoral axis and the extension of the trochlear line, which was used to measure TL (Fig 1). The proximal and distal patellar positions (PPP and DPP, respectively) were measured on the extension of the trochlear line and defined as the lengths from the proximal end of the trochlea to the distal direction divided by TL, as described in a previous study (Fig 3) [8].

All measurements were done by one person (SM) and performed with computer-aided design software (AR_CAD v1.6.0; SHF Co., Kyoto, Japan).

The ratios of the measured lengths were used instead of the measured lengths themselves. The ratios evaluated were the PLL/PL, FC/PL, TL/PL, and TL/FC.

## Statistical analyses

First, simple linear regression analyses were performed with all the radiographs to evaluate the association between the PPP or DPP and the joint angle, sex, limb side, body weight, or age. Variables with a p-value < 0.20 were included in multivariable models [18, 19]. Multivariable regression analyses were then performed to assess the association between the PPP or DPP

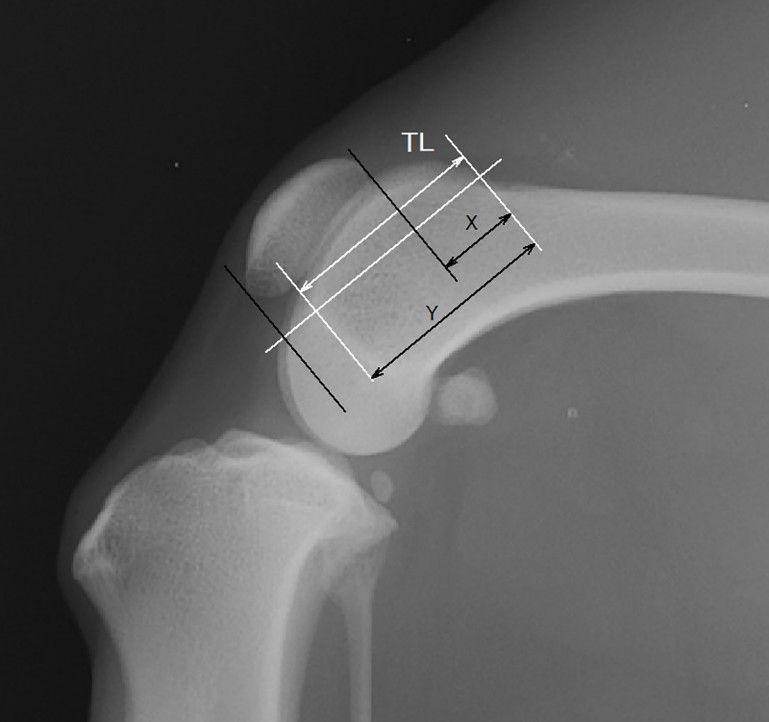

**Fig 3. The measurement definitions of the femoral trochlear length (TL), proximal patellar position (PPP), and distal patellar position (DPP).** The TL was measured from the proximal extent of the femoral trochlear ridges to the origin of the long digital extensor muscle. X is the distance from the proximal end of the TL to the proximal edge of the patella. The PPP was defined as X/TL. Y is the distance from the proximal end of the TL to the distal edge of the patella. The DPP was defined as Y/TL.

and the PLL/PL, AT angle, FC/PL, TL/PL, or TL/FC. Subsequently, multivariable regression analyses were also performed on the PPP or DPP and the presence of MPL. If the presence of MPL had a p-value < 0.10 in the final model, multivariable regression analyses with MPL grade as an independent variable were subsequently performed.

Simple linear regression analyses were performed to evaluate the association between PLL/PL, AT angle, FC/PL, TL/PL, or TL/FC, and the joint angle, sex, limb side, body weight, or age. Variables with a p-value < 0.20 were included in multivariable models [18, 19]. Multivariable regression analyses were then performed to assess the association between PLL/PL, AT angle, FC/PL, TL/PL, or TL/FC and the presence of MPL. If the presence of MPL had a p-value < 0.10 in the final model, multivariable regression analyses with MPL grade as an independent variable were subsequently performed.

For multivariable regression analyses, multiple linear regressions with backward elimination were performed to identify a model containing variables whose coefficients significantly differed from 0. Collinearity between variables was checked by the variance inflation factor. Variables with the variance inflation factor exceeding 4.0 were excluded from the final model. Interactions between variables were checked by performing likelihood ratio tests contrasting models with and without the interactions. Any clinically important or statistically significant interaction was included in the final model. Stata (version 14, StataCorp, College Station, TX) was used for all analyses. For statistical estimation and inferences, two-sided hypothesis tests were used with a 5% significance level.

Intra-rater and inter-rater reliability of the stifle joint angle, PPP, DPP, PLL/PL, AT angle, FC/PL, TL/PL, and TL/FC were measured by using ten radiographs that were randomly selected from the study population. One observer (SM) measured each value three times on different days, and the intra-rater reliability was analysed using a one-way random effect model. Then, three observers (SM, MS, and one student) measured each value once, and the inter-rater reliability was analysed using a two-way model with absolute agreement. The intra-class correlation coefficient for each value was calculated using SPSS (version 27, IBM Corp, Armonk, NY).

## Results

Overall, radiographs of 99 limbs of 71 dogs belonging to 20 breeds were assessed in this study, including 21 Toy Poodles, 9 mixed breeds, 7 Chihuahuas, 6 Yorkshire Terriers, 5 Pomeranians, 3 Jack Russel Terriers, 3 Papillons, 3 Welsh Corgis, 2 Border Collies, 2 Shiba Inus, 1 Beagle, 1 Cavalier King Charles Spaniel, 1 Chin, 1 English Cocker Spaniel, 1 French Bulldog, 1 Maltese, 1 Miniature Dachshund, 1 Miniature Schnauzer, 1 Shetland Sheepdog, and 1 Shih Tzu. There were 10 intact males, 10 intact females, 27 castrated males, and 24 spayed females with a median body weight of 4.15 kg (range: 1.15–14.2 kg) and a median age of 65.3 months (range: 6.8–173.5 months). Of the 99 limbs in this study, 53 were left limbs and 46 were right limbs. These included 36 limbs with no patellar luxation, 9 with grade 1 MPL, 25 with grade 2 MPL, and 29 with grade 3 MPL. Table 1 shows the breed, sex, median body weight, and median age of the dogs in the MPL and control groups. The mean values of the measured ratios and angles of the MPL and control groups are shown in Table 2.

Table 3 shows the results of the simple linear regression analyses for PPP and DPP. Based on these results, joint angle, body weight, and age were included in each multivariable model analysis. Table 4 shows the results of the multivariable regression analyses for PPP or DPP and morphological factors. Both final models included joint angle, age, PLL/PL, AT angle, TL/PL, and TL/FC. Table 5 shows the results of the multivariable regression analyses for PPP or DPP and the presence of MPL. Both final models revealed that PPP and DPP were not significantly different between the MPL and control groups, although the p-value for MPL in the final model for DPP was 0.073. Similar results were obtained when the analysis was limited to the toy poodle breed (S3 Table). The final models of the multivariable regression analyses for PPP or DPP and the presence of MPL revealed that PPP and DPP were not significantly different between the MPL and control groups in toy poodles. Results of the multivariable regression analyses for DPP and MPL grade are shown in Table 6. Hindlimbs with grade 3 MPL had a trend toward greater DPP (p = 0.079).

Table 7 shows the results of the multivariable regression analyses for PLL/PL, AT angle, FC/PL, TL/PL, or TL/FC, and the presence of MPL. The PLL/PL and TL/FC were not significantly different between the MPL and control groups. The AT angle was significantly bigger in the MPL group compared with the control group. FC/PL and TL/PL were significantly smaller in the MPL group compared with the control group. For AT angle, FC/PL, and TL/PL, multivariable regression analyses were also performed to evaluate the association with MPL grade (Table 8). Grade 2 (p = 0.053) and 3 (p = 0.087) displayed a trend toward a greater AT angle than that of the control group, but the differences were not significant. FC/PL and TL/PL were significantly smaller in grade 2 (p = 0.007 for FC/PL, p = 0.026 for TL/PL) and 3 (p < 0.001 for both) compared to those of the control group. AT angle, FC/PL and TL/PL were not significantly different in grade 1 compared to those of the control group.

All variables had a variance inflation factor less than 4.0 in all final models (S2 Table). No interaction between variables was statistically significant.

**Table 1.  The breed, sex, median body weight (range), and median age (range) of the dogs in the medial patellar luxation (MPL) and control groups.**

| | | MPL group | Control group | Total |
|---|---|---|---|---|
| **Breed** | Toy Poodles | 13 | 8 | 21 |
| | Mixed breeds | 7 | 2 | 9 |
| | Chihuahuas | 5 | 2 | 7 |
| | Yorkshire Terriers | 5 | 1 | 6 |
| | Pomeranians | 5 | - | 5 |
| | Jack Russel Terriers | - | 3 | 3 |
| | Papillons | 1 | 2 | 3 |
| | Welsh Corgis | - | 3 | 3 |
| | Border Collies | - | 2 | 2 |
| | Shiba Inus | 2 | - | 2 |
| | Beagle | - | 1 | 1 |
| | Cavalier King Charles Spaniel | 1 | - | 1 |
| | Chin | 1 | - | 1 |
| | English Cocker Spaniel | 1 | - | 1 |
| | French Bulldog | - | 1 | 1 |
| | Maltese | 1 | - | 1 |
| | Miniature Dachshund | 1 | - | 1 |
| | Miniature Schnauzer | - | 1 | 1 |
| | Shetland Sheepdog | 1 | - | 1 |
| | Shih Tzu | - | 1 | 1 |
| **Sex** | Male | 6 | 4 | 10 |
| | Female | 6 | 4 | 10 |
| | Castrated male | 14 | 13 | 27 |
| | Spayed female | 18 | 6 | 24 |
| **Body weight (kg)** | Median (range) | 3.70 (1.15–11.3) | 4.58 (1.60–14.2) | 4.15 (1.15–14.2) |
| **Age (months)** | Median (range) | 53.1 (7.4–139.8) | 97.9 (6.8–173.5) | 65.3 (6.8–173.5) |

Abbreviations: MPL, medial patellar luxation.

Intra-rater reliability was almost perfect for all values as intraclass correlation coefficients for repeated measurement of one observer (SM) on the stifle joint angle, PPP, DPP, PLL/PL, AT angle, FC/PL, TL/PL, and TL/FC were 0.99, 0.97, 0.96, 0.90, 0.91, 0.83, 0.83, and 0.88,

**Table 2.  Mean values (standard deviations) of the measurement ratios and angles of the MPL and control groups.**

| | MPL group | Control group |
|---|---|---|
| **Stifle joint angle (degrees)** | 95 (± 12) | 81 (± 13) |
| **PPP** | 0.28 (± 0.18) | 0.46 (± 0.19) |
| **DPP** | 0.99 (± 0.16) | 1.10 (± 0.17) |
| **PLL/PL** | 1.87 (± 0.23) | 1.93 (± 0.21) |
| **AT angle (degrees)** | 137 (± 6) | 135 (± 5) |
| **FC/PL** | 0.90 (± 0.07) | 0.95 (± 0.10) |
| **TL/PL** | 1.43 (± 0.16) | 1.55 (± 0.13) |
| **TL/FC** | 1.59 (± 0.19) | 1.63 (± 0.17) |

Abbreviations: AT angle, anatomical trochlear angle; DPP, distal patellar position; FC, craniocaudal size of the femoral condyle; PL, patellar length; PLL, patellar ligament length; PPP, proximal patellar position; TL, femoral trochlear length.

**Table 3. Results of simple linear regression analyses for PPP or DPP and stifle joint angle, sex, age, body weight, or limb side.**

| PPP | | Coefficient | 95% CI | p | Adj. $R^2$ |
|---|---|---|---|---|---|
| Joint angle | | -0.0112 | -0.0130–(-0.00937) | <0.001 | 0.601 |
| Cons | | 1.36 | 1.19–1.52 | <0.001 | |
| Sex | Spayed | -0.0412 | -0.141–0.0589 | 0.416 | -0.0184 |
| | | -0.0306 | -0.154–0.0926 | 0.623 | |
| | Male | -0.0615 | -0.182–0.0592 | 0.314 | |
| Cons | Female | 0.376 | 0.307–0.446 | <0.001 | |
| Limb side | Right | -0.0264 | -0.108–0.0550 | 0.521 | -0.0060 |
| Cons | | 0.360 | 0.304–0.415 | <0.001 | |
| Body weight | | 0.0160 | 0.00381–0.0281 | 0.011 | 0.0558 |
| Cons | | 0.270 | 0.199–0.341 | <0.001 | |
| Age | | 0.00159 | 0.000829–0.00235 | <0.001 | 0.142 |
| Cons | | 0.244 | 0.181–0.306 | <0.001 | |
| DPP | | Coefficient | 95% CI | p | Adj $R^2$ |
| Joint angle | | -0.00923 | -0.0108–(-0.00761) | <0.001 | 0.565 |
| Cons | | 1.87 | 1.72–2.02 | <0.001 | |
| Sex | Spayed | -0.0266 | -0.112–0.0586 | 0.537 | -0.0238 |
| | Male | -0.0347 | -0.140–0.0702 | 0.513 | |
| | Female | -0.0341 | -0.137–0.0687 | 0.512 | |
| Cons | | 1.05 | 0.995–1.11 | <0.001 | |
| Limb side | Right | -0.0132 | -0.0825–0.0560 | 0.705 | -0.0088 |
| Cons | | 1.04 | 0.993–1.09 | <0.001 | |
| Body weight | | 0.0116 | 0.00122–0.0221 | 0.029 | 0.0384 |
| Cons | | 0.977 | 0.917–1.04 | <0.001 | |
| Age | | 0.00114 | 0.000478–0.00180 | 0.001 | 0.0981 |
| Cons | | 0.960 | 0.905–1.01 | <0.001 | |

For sex, spayed females, male, and female were compared with castrated males.

Limb side was the right side compared with the left side.

Adj., adjusted; CI, confidence interval; Cons, constant; DPP, distal patellar position; PPP, proximal patellar position.

respectively (p<0.01). Inter-rater reliability of these values was substantial to almost perfect as the intraclass correlation coefficients for the three observers (SM, MS, and one student) on the stifle joint angle, PPP, DPP, PLL/PL, AT angle, FC/PL, TL/PL, and TL/FC were 0.98, 0.95, 0.92, 0.92, 0.86, 0.84, 0.73, and 0.80, respectively (p<0.01).

## Discussion

Consistent with our hypothesis, the proximodistal patellar position was dependent on multiple factors. A smaller PLL/PL, greater AT angle, greater TL/PL, or smaller TL/FC could lead to a distally positioned patella. Although the TL/PL of the MPL group was significantly smaller than that of the control group, the second hypothesis could not be confirmed because the proximodistal patellar position in the MPL group was not significantly different from that in the control group. This might be because the AT angle in the MPL group was significantly greater than that in the control group.

The stifle joint angle result showed that as the stifle joint angle became smaller (stifle joint flexed), the proximodistal patellar position became more distal. The coefficients of the joint angle for PPP and DPP were similar in all models, suggesting that the influence of the stifle joint angle on the proximodistal patellar position was barely affected by the other variables.

**Table 4. Final models of the multivariable regression analyses for PPP or DPP and morphological factors.**

| PPP | Coefficient | 95% CI | P | Adj. $R^2$ |
|---|---|---|---|---|
| **Joint angle** | -0.0102 | -0.0113–(-0.00905) | <0.001 | 0.863 |
| **Age** | 0.000561 | 0.000227–0.000894 | 0.001 | |
| **PLL/PL** | -0.359 | -0.435–(-0.283) | <0.001 | |
| **AT angle** | 0.00641 | 0.00387–0.00896 | <0.001 | |
| **TL/PL** | 0.659 | 0.517–0.800 | <0.001 | |
| **TL/FC** | -0.197 | -0.315–(-0.0797) | 0.001 | |
| **Cons** | 0.382 | -0.0470–0.812 | 0.080 | |
| **DPP** | **Coefficient** | **95% CI** | **p** | **Adj. $R^2$** |
| **Joint angle** | -0.00966 | -0.0108–(-0.00854) | <0.001 | 0.817 |
| **Age** | 0.000544 | 0.000216–0.000872 | 0.001 | |
| **PLL/PL** | -0.340 | -0.414–(-0.265) | <0.001 | |
| **AT angle** | 0.00696 | 0.00446–0.00946 | <0.001 | |
| **TL/PL** | 0.200 | 0.0606–0.339 | 0.005 | |
| **TL/FC** | -0.205 | -0.321–(-0.0893) | 0.001 | |
| **Cons** | 1.60 | 1.18–2.02 | <0.01 | |

Abbreviations: Adj., adjusted; AT angle, anatomical trochlear angle; CI, confidence interval; Cons, constant; DPP, distal patellar position; FC, craniocaudal size of the femoral condyle; PL, patellar length; PLL, patellar ligament length; PPP, proximal patellar position; TL, femoral trochlear length.

The coefficients of the joint angle in all final models for the PPP and DPP were approximately -0.01, suggesting that both the PPP and DPP were similarly influenced by the joint angle. These results are consistent with those of a previous study on 13 dogs, although that study did not report the actual coefficients [8].

The PLL/PL would offer some benefit in the assessment of the proximodistal patellar position since it was included in the final models for both PPP and DPP. However, assessment of the proximodistal patellar position based on PLL/PL alone may not be feasible, since other values were also included in the final models. The effects of AT angle and TL/PL on the proximodistal patellar position imply that a trochlea near-vertical to the long axis of femur or a shorter trochlea would lead to a proximally positioned patella. The results of TL/FC on PPP and DPP indicate that when the other values were considered, the patella was positioned proximally as the FC became smaller. A smaller femoral condyle arc might lead to a distally positioned

**Table 5. Final models of multivariable linear regression analyses for PPP or DPP and the presence of MPL.**

| PPP | Coefficient | 95% CI | p | Adj. $R^2$ |
|---|---|---|---|---|
| **Joint angle** | -0.0106 | -0.0125–(-0.00859) | <0.001 | 0.644 |
| **Age** | 0.000947 | 0.000427–0.00147 | <0.001 | |
| **MPL** | 0.00609 | -0.0531–0.0653 | 0.839 | |
| **Cons** | 1.24 | 1.06–1.41 | <0.001 | |
| **DPP** | **Coefficient** | **95% CI** | **p** | **Adj. $R^2$** |
| **Joint angle** | -0.00950 | -0.0113–(-0.00772) | <0.001 | 0.598 |
| **Age** | 0.000700 | 0.000230–0.00117 | 0.004 | |
| **MPL** | 0.0489 | -0.00460–0.102 | 0.073 | |
| **Cons** | 1.81 | 1.66–1.97 | <0.001 | |

MPL signifies the MPL group compared with the control group.

Adj., adjusted; CI, confidence interval; Cons, constant; DPP, distal patellar position; MPL, medial patellar luxation; PPP, proximal patellar position.

**Table 6. Final models of multivariable linear regression analyses for DPP and the grade of MPL.**

| DPP | | Coefficient | 95% CI | p | Adj. R$^2$ |
|---|---|---|---|---|---|
| **Joint angle** | | -0.00953 | -0.0113–(-0.00772) | <0.001 | 0.590 |
| **Age** | | 0.000712 | 0.000232–0.00119 | 0.004 | |
| **MPL** | Grade 1 | 0.0387 | -0.0448–0.122 | 0.360 | |
| | Grade 2 | 0.0460 | -0.0193–0.111 | 0.165 | |
| | Grade 3 | 0.0563 | -0.00654–0.119 | 0.079 | |
| **Cons** | | 1.82 | 1.66–1.98 | <0.001 | |

Each MPL grade was compared with the control group.

Abbreviations: Adj., adjusted; CI, confidence interval; Cons, constant; DPP, distal patellar position; MPL, medial patellar luxation

trochlea and thus a proximally positioned patella. Age was included in the final models for both PPP and DPP. However, its coefficients were small, and more than 17 months of age would be required to alter the PPP or DPP by 1%. Thus, the clinical importance of age for the proximodistal patellar position might be limited.

The PLL/PL of the MPL group was not greater than that of the control group. This was consistent with the findings of previous studies with specific small-breed dogs that did not show a significant difference in the PLL/PL between dogs with and without MPL [4, 17]. Although the anatomical features, as well as PLL/PL, had a significant influence on the proximodistal patellar position, PPP and DPP were not significantly different between the MPL and control groups. While the smaller TL/PL of the MPL group could be related to a proximally positioned patella, the smaller AT angle of the MPL group might have counteracted the effect of TL/PL. Interestingly, the MPL group tended to show a greater DPP in the final model (p = 0.073).

**Table 7. Final models of multivariable linear regression analyses for PLL/PL, AT angle, FC/PL, TL/PL, or TL/FC, and the presence of MPL.**

| PLL/PL | Coefficient | 95% CI | P | Adj. R$^2$ |
|---|---|---|---|---|
| **MPL** | -0.0581 | -0.152–0.0363 | 0.225 | 0.0050 |
| **Cons** | 1.93 | 1.85–2.00 | <0.001 | |
| **AT angle** | **Coefficient** | **95% CI** | **p** | **Adj. R$^2$** |
| **Body weight** | 0.433 | 0.0606–0.805 | 0.023 | 0.0598 |
| **MPL** | 2.80 | 0.296–5.30 | 0.029 | |
| **Cons** | 132 | 130–135 | <0.001 | |
| **FC/PL** | **Coefficient** | **95% CI** | **p** | **Adj. R$^2$** |
| **Age** | -0.000476 | -0.000830–(-0.000122) | 0.009 | 0.129 |
| **MPL** | -0.0688 | -0.105–(-0.0325) | <0.001 | |
| **Cons** | 0.994 | 0.953–1.04 | <0.001 | |
| **TL/PL** | **Coefficient** | **95% CI** | **p** | **Adj. R$^2$** |
| **MPL** | -0.119 | -0.181–(-0.563) | <0.001 | 0.119 |
| **Cons** | 1.55 | 1.50–1.60 | <0.001 | |
| **TL/FC** | **Coefficient** | **95% CI** | **p** | **Adj. R$^2$** |
| **Age** | 0.00130 | 0.000560–0.00204 | <0.001 | 0.105 |
| **MPL** | 0.000983 | -0.0748–0.07768 | 0.980 | |
| **Cons** | 1.52 | 1.44–1.61 | <0.001 | |

MPL signifies the MPL group compared with the control group.

Abbreviations: Adj., adjusted; AT angle, anatomical trochlear angle; CI, confidence interval; Cons, constant; DPP, distal patellar position; FC, craniocaudal size of the femoral condyle; MPL, medial patellar luxation; PL, patellar length; PLL, patellar ligament length; PPP, proximal patellar position; TL, femoral trochlear length.

**Table 8. Final models of multivariable linear regression analyses for AT angle, FC/PL, TL/PL or TL/FC, and the grade of MPL.**

| AT angle | | Coefficient | 95% CI | p | Adj. $R^2$ |
|---|---|---|---|---|---|
| Body weight | | 0.431 | 0.0509–0.811 | 0.027 | 0.0407 |
| MPL | Grade 1 | 2.55 | -1.83–6.92 | 0.251 | |
| | Grade 2 | 3.05 | -0.0450–6.15 | 0.053 | |
| | Grade 3 | 2.65 | -0.396–5.69 | 0.087 | |
| Cons | | 132 | 130–135 | <0.001 | |
| FC/PL | | Coefficient | 95% CI | p | Adj. $R^2$ |
| Age | | -0.000501 | -0.000862–(-0.000140) | 0.007 | 0.124 |
| MPL | Grade 1 | -0.0486 | -0.110–0.0129 | 0.120 | |
| | Grade 2 | -0.0625 | -0.108–(-0.0176) | 0.007 | |
| | Grade 3 | -0.0822 | -0.125–(-0.0389) | <0.001 | |
| Cons | | 0.997 | 0.955–1.04 | <0.001 | |
| TL/PL | | Coefficient | 95% CI | p | Adj. $R^2$ |
| MPL | Grade 1 | -0.0617 | -0.171–0.0478 | 0.266 | 0.147 |
| | Grade 2 | -0.0872 | -0.164–(-0.0107) | 0.026 | |
| | Grade 3 | -0.163 | -0.237–(-0.0901) | <0.001 | |
| Cons | | 1.55 | 1.50–1.60 | <0.001 | |

Each MPL grade was compared with the control group.

Abbreviations: Adj., adjusted; AT angle, anatomical trochlear angle; CI, confidence interval; Cons, constant; FC, craniocaudal size of the femoral condyle; MPL, medial patellar luxation; PL, patellar length; PLL, patellar ligament length; TL, femoral trochlear length.

This trend was also seen in grade 3 MPL when compared with the control group (p = 0.079). Although a distally positioned patella has been mentioned in relation to lateral patellar luxation [5, 7], it might also plausibly be a risk factor for MPL, since exceeding the femoral trochlea distally could increase the instability both medially and laterally.

The significantly smaller TL/PL in the MPL group indicated the presence of trochlear dysplasia in the MPL group. Although previous studies identified the shallow femoral trochlear groove as a consequence of the lack of proper articulation between the patella and the trochlear groove in dogs with MPL [20–22], our result suggests that trochlear length might also be reduced in the absence of proper articulation. The result of smaller TL/PL with a higher MPL grade may also support this theory because the proper articulation of the patella and trochlear groove is less likely where MPL grade is higher [21, 22]. Similarly, hypoplasia of the femoral condyle, especially the medial femoral condyle, in dogs with MPL has been mentioned in previous studies [20–22]. The smaller FC/PL of the MPL group compared with that of the control group might reflect the hypoplasia of the femoral condyle. FC/PL was also smaller with higher MPL grade, which might indicate greater severity of hypoplasia of the femoral condyle with a higher MPL grade.

This study has certain limitations. First, no dog breed other than the toy poodle was analysed in this study. Thus, some of the findings for MPL in this study could be attributable to the differences between breeds predisposed to MPL and those not showing the predisposition. To investigate the intra-breed validity of our findings for breeds other than the toy poodle, breed-specific studies are required. Another limitation was that the range of motion of the patella was not considered in this study. Hyperextension of the stifle joint would lead to a proximally positioned patella even with a normal proximodistal patellar position in relation to the stifle joint angle. To assess the range of motion of the patella, additional studies are warranted.

In conclusion, the proximodistal patellar position of small dogs was dependent on the size of the femoral condyle and the shape of the trochlea, as well as the length of the patellar

ligament. A shorter patellar ligament, an upright trochlea, greater trochlear length, and bigger femoral condyle will lead to a distally positioned patella. Furthermore, the proximodistal patellar position was not significantly different between the hindlimbs with MPL and those without patellar luxation. Although hindlimbs with MPL showed a significantly shorter trochlea than those without patellar luxation, this difference did not appear to be sufficient to displace the patellar position proximally in small dogs.

## Supporting information

**S1 File. Details and measurements obtained for the 71 cases.**
(XLSX)

**S1 Table. Results of simple linear regression analyses for PLL/PL, AT angle, FC/PL, TL/PL, or TL/FC, and stifle joint angle, sex, age, body weight, or limb side.**
(DOCX)

**S2 Table. The variance inflation factors of all the independent variables in the final models of multiple linear regression.**
(DOCX)

**S3 Table. Final models of multivariable linear regression analyses for PPP or DPP and the presence of MPL in toy poodles.**
(DOCX)

**S4 Table. Measurement values obtained by each observer (three times by SM and once each by MS and the student) for 10 cases.**
(DOCX)

## Acknowledgments

We would like to thank Aki Tanaka, DVM, PhD, from the Department of Wild Animals at Nippon Veterinary and Life Science University for beneficial advice on the statistical analyses used in this manuscript.

We would also like to thank Daisuke Minagawa, DVM, and Mariko Ozawa, DVM, of Minagawa Pet Clinic for providing radiographs of clinical cases.

We would also like to thank Editage (www.editage.com) for English language editing.

## Author Contributions

**Conceptualization:** Sawako Murakami.

**Data curation:** Sawako Murakami, Masakazu Shimada.

**Formal analysis:** Sawako Murakami, Masakazu Shimada.

**Investigation:** Sawako Murakami, Masakazu Shimada.

**Methodology:** Sawako Murakami, Masakazu Shimada.

**Resources:** Yasuji Harada, Yasushi Hara.

**Supervision:** Yasuji Harada, Yasushi Hara.

**Validation:** Yasuji Harada, Yasushi Hara.

**Writing – original draft:** Sawako Murakami.

**Writing – review & editing:** Sawako Murakami, Yasuji Harada, Yasushi Hara.

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
