## [Decision Letter · Decision Letter 0]

2 Mar 2021

PONE-D-20-37854

Examination of the proximodistal patellar position in small dogs in relation to anatomical features of the distal femur and medial patellar luxation

PLOS ONE

Dear Dr. Murakami,

Thank you for submitting your manuscript to PLOS ONE. After careful consideration, we feel that it has merit but does not fully meet PLOS ONE’s publication criteria as it currently stands. Therefore, we invite you to submit a revised version of the manuscript that addresses the points raised during the review process.

We look forward to receiving your revised manuscript.

Kind regards,

Silvia Sabattini

Academic Editor

PLOS ONE

Journal Requirements:

Reviewers' comments:

Reviewer's Responses to Questions

**Comments to the Author**

1. Is the manuscript technically sound, and do the data support the conclusions?

Reviewer #1: Yes

Reviewer #2: Partly

Reviewer #3: Yes

2. Has the statistical analysis been performed appropriately and rigorously? 

Reviewer #1: I Don't Know

Reviewer #2: I Don't Know

Reviewer #3: I Don't Know

3. Have the authors made all data underlying the findings in their manuscript fully available?

Reviewer #1: Yes

Reviewer #2: No

Reviewer #3: No

4. Is the manuscript presented in an intelligible fashion and written in standard English?

Reviewer #1: Yes

Reviewer #2: Yes

Reviewer #3: Yes

5. Review Comments to the Author

Reviewer #1: Comments to the Authors

Maniscuript PONE-D-20-37854 “Examination of the proximodistal patellar position in small dogs in relation to anatomical features of the distal femur and medial patellar luxation” is concisely and presents information of importance to determine the influence of anatomical features of the distal femur on theproximodistal patellar position and compare the proximodistal patellar position between dogs with and without medial patellar luxation (MPL).

Specific comments are presented below.

Line 51-52: (Introduction) please explain “patellar height index”. Insall-Salvati index is explained above, but patellar height index is not.

Line 88-89: (Material and Methods) Please explain “How did you determine the MPL grade in radiography?”. you can give literature or explain.

Line 94-108.(Measurements). Since there are morphometric measurements in the study, it would be good to explain some things. Did you make a CV before starting the measurements for the reliability of the measurements. And please specify if measurements are made by one person.

Line 96: (Measurements). Anatomical axes of the distal femur determineted according to reference 5. But it is different from reference 5 in the fig. 1. The distance between FW and D used to determine is 20mm this axis in the reference 5. In this article is B/2. ( B: femoral condyle length.). Please explain if B = 20mm.If FW and D are not defined correctly, axis and angle will be wrong.

Line 101. Please explain “extension of the trochlear line”. Please indicate if you used to reference.

Line 110: Fig 1. Please revise it. Prefer standard usages (eg. FW, D, FCL) for clarity.

Line 172-177 . (Result). Please explain “animal numbers specified do not match S1 Table. Details of dog breeds for each MPL grade. Please check again.

Line 186: (Table 1). The breeds part in the table is very mixed. It will be clearer as follows.

Suggested table:

Table 1

Breed MPL group Control group Total

Toy Poodles 13 8 21

Mixed breeds 7 2 9

Chihuahuas 5 2 7

Pomeranians 5 - 5

Yorkshire Terriers 5 1 6

Shiba Inus 2 - 2

Cavalier King Charles Spaniel 1 - 1

Chin 1 - 1

English Cocker Spaniel 1 - 1

Maltese 1 - 1

Miniature Dachshund 1 - 1

Papillon 1 2 3

Shetland Sheepdog 1 - 1

Jack Russel Terriers - 3 3

Welsh Corgis - 3 3

Border Collies - 2 2

Beagle - 1 1

French Bulldog - 1 1

Miniature Schnauzer - 1 1

Shih Tzu - 1 1

44 27 71

Line 190. (Table 2). Please change all numbers with decimals to avoid confusion. (95= 95.00)

Line 241(Result): Please check MPL grade? (Is it grade1?)

S3 Table: Please check the accuracy..Two tables are the same (PPP and DPP).

Reviewer #2: Dear. the author

The authors evaluate Examination of the proximodistal patellar position in small dogs in relation to anatomical features of the distal femur and medial patellar luxation.

The reviewer suggests rejecting the article first of all because to gross methodological errors.

Kind regards,

Reviewer #3: The key findings in this study are

1) The proximodistal patellar position of small dogs depends on the size of the femoral condyle, the shape of the trochlea, and the length of patellar ligament.

2) A shorter patellar ligament, an upright trochlea, greater trochlear length, and bigger femoral condyle will lead to a distally positioned patella.

3) The proximodistal patellar position is not significantly different between the hindlimbs with and without MPL.

The main limitation with this study is that the authors included 20 dog breeds in their study and due to the fact that there are differences between the dog breeds regarding the measurements mentioned in this study, it will be difficult to get precise results.

Another limitation is that no consideration is given to the fact that flexion and extension of the joint can affect the position of the patella within the femoral trochlea. In this case, the joint angle measurement is irrelevant and should not be taken into account.

General Comment:

Where possible, avoid writing in the passive voice.

Rephrase sentences to be to the point. Avoid dragging out sentences by making it too wordy.

The overuse of prepositions/non-content words makes sentences hard to follow and unclear to understand. Consider rewriting/rephrasing where needed.

34 Insert a comma before “and”

62 Consider replacing “On the basis of” with “Based on”

66 Replace “and” with “or”

68 Replace “and” with “or”

70 – 77 Change the tense of the objective and hypothesis to present tense (change “was” to “is”)

80 Consider removing “that were”

85 The meaning of the sentence might be unclear. Consider using “or” instead of “and.” Alternatively, consider using a comma before “and”

90-92 Consider rewriting the sentence, using fewer non-content words.

95 Consider rewriting the sentence starting with: “Based on a previous study…”

111-112 The values of the joint angle cannot be constant in all measured x-rays as it depends on the flexion and extension position of the joint. It should not be taken into account in this study

112-113 To ensure accurate anatomical femoral axis, you need to take the whole femoral shaft into consideration. See Aiyan et al. “Measurement of the femoral neck angle in medium and large dog breeds” Acta Veterinaria Hungarica 67 (1), pp. 22–33 (2019). DOI: 10.1556/004.2019.003

113 Consider labelling the two points (white squares).

113 I think you mean the “distal point” not “proximal point”.

115 I think you mean the “proximal point” not “distal point”.

116 “distal point”. The “proximal” tibial anatomical axis

110-121 General comment regarding Fig.1: I suggest, providing a detailed description of the method used to identify the anatomical axis of the femur and tibia in the M&M and provide the references for all measurement methods. In this case, include only the labels in the figure legend

123 Fig.2 you already mention the definitions in the M&M (avoid repetition)

130-135 The distal white line in fig. 3 is located few mm below/distal to the origin of the long digital extensor muscle! Provide references for the calculation of the PPP and DPP.

142 As mentioned previously, the joint angle depends on the flexion and extension position of the joint and it cannot be constant in all studied dogs. (I think, you have to exclude it from your statistical analyses)

152 This sentence might be unclear, consider rephrasing it. Ensure the correct use of “or” vs “and”

175 Consider rewriting the sentence. Remain consistent: “1 Beagle, 1 Cavalier King…” or “a Beagle, a Cavalier King…”

180 Insert a comma before “and”

191 Including the stifle joint angle will confuse the readers for the same reason mentioned previously. See the large SD range

210 Consider rephrasing this sentence.

223 The phrase “control group” is missing a determiner: “a” or “the”

233 The sentence is missing a full stop at the end.

239 Change “were” to “was”

250 The phrase “control group” is missing a determiner: “a” or “the”

256 “variance” is missing a determiner: “a”

279 Consider rephrasing to: “the long femoral axis”

280 Consider rephrasing to: “The TL/FC results…”

289 Insert a comma before and after the interruption: “…features, as well as PLL/PL, had…”

307 Consider removing the word: “some”

310 Remove the word: “a”

314 Remove the word: “such”

323 The phrase “patellar ligament” is missing a determiner: “the patellar ligament.”

6. PLOS authors have the option to publish the peer review history of their article (what does this mean?). If published, this will include your full peer review and any attached files.

Reviewer #1: No

Reviewer #2: No

Reviewer #3: **Yes: **Ahmad Al Aiyan

---

## [Author Response · Author response to Decision Letter 0]

10 Apr 2021

Reviewer #1: Comments to the Authors

Maniscuript PONE-D-20-37854 “Examination of the proximodistal patellar position in small dogs in relation to anatomical features of the distal femur and medial patellar luxation” is concisely and presents information of importance to determine the influence of anatomical features of the distal femur on the proximodistal patellar position and compare the proximodistal patellar position between dogs with and without medial patellar luxation (MPL).

Specific comments are presented below.

Line 51-52: (Introduction) please explain “patellar height index”. Insall-Salvati index is explained above, but patellar height index is not.

Response: Thank you for the comment. I had misused the word “patellar height index”; thus, I rephrased it to “the cartilage congruence of patellofemoral joint measured on MRI”. Please check lines 56-59.

Line 88-89: (Material and Methods) Please explain “How did you determine the MPL grade in radiography?”. you can give literature or explain.

Response: I apologize for the confusion. The MPL grade was obtained from the patient records. I added this information on lines 91-92 as “Breed, sex, age, body weight, limb side, and grade of MPL were obtained from the patient records.”

Line 94-108.(Measurements). Since there are morphometric measurements in the study, it would be good to explain some things. Did you make a CV before starting the measurements for the reliability of the measurements. And please specify if measurements are made by one person.

Response: I calculated the interclass correlation coefficient of the measurements to determine the intra-rater and inter-rater reliability. I added this method on lines 184-191 and the results on lines 291-298.

I also added “All measurements were done by one person (SM)” on line 151.

Line 96: (Measurements). Anatomical axes of the distal femur determineted according to reference 5. But it is different from reference 5 in the fig. 1. The distance between FW and D used to determine is 20mm this axis in the reference 5. In this article is B/2. ( B: femoral condyle length.). Please explain if B = 20mm.If FW and D are not defined correctly, axis and angle will be wrong.

Response: Anatomical axes were determined based on reference no. 5. However, I made several changes because the dogs in the current study were smaller than those in reference no. 5. I used B/2 instead of 20 mm and 1.5*A instead of 2*PTW because 20 mm or 2*PTW were too large for small dogs. These changes were made to plot the distal femoral axis and proximal tibial axis in small dogs. Please check lines 100-102.

Line 101. Please explain “extension of the trochlear line”. Please indicate if you used to reference. 

Response: I changed the order of the sentences and moved the explanation about AT angle after the explanation of TL. Please check lines 116-118: “The anatomical trochlear angle (AT angle) was defined as the caudal angle between the distal femoral axis and the extension of the trochlear line, which was used to measure TL.”

Line 110: Fig 1. Please revise it. Prefer standard usages (eg. FW, D, FCL) for clarity.

Response: As advised, I revised the Figure legend and used FCL and PTW accordingly.

Line 172-177 . (Result). Please explain “animal numbers specified do not match S1 Table. Details of dog breeds for each MPL grade. Please check again.

Response: I apologize for the confusion. I miswrote the number of dogs in the previous manuscript, and it was corrected accordingly.

Line 186: (Table 1). The breeds part in the table is very mixed. It will be clearer as follows.

Suggested table:

Table 1

Breed MPL group Control group Total

Toy Poodles 13 8 21

Mixed breeds 7 2 9

Chihuahuas 5 2 7

Pomeranians 5 - 5

Yorkshire Terriers 5 1 6

Shiba Inus 2 - 2

Cavalier King Charles Spaniel 1 - 1

Chin 1 - 1

English Cocker Spaniel 1 - 1

Maltese 1 - 1

Miniature Dachshund 1 - 1

Papillon 1 2 3

Shetland Sheepdog 1 - 1

Jack Russel Terriers - 3 3

Welsh Corgis - 3 3

Border Collies - 2 2

Beagle - 1 1

French Bulldog - 1 1

Miniature Schnauzer - 1 1

Shih Tzu - 1 1

44 27 71

Response: Thank you for your helpful suggestion. I revised Table 1 as per your advice. I also deleted the limb side information because it was about the number of limbs, not the number of dogs as for the other values in this table.

Line 190. (Table 2). Please change all numbers with decimals to avoid confusion. (95= 95.00)

Response: I apologize for the confusion. The angles were direct measurements from the software, and the ratios were calculated from the length measured in the software. Hence, the number of significant figures differs. I used integers for the angles, and ratios were rounded off to two decimal places.

Line 241(Result): Please check MPL grade? (Is it grade1?)

Response: I beg your pardon; I did not quite catch what you meant to say. The multivariable regression analyses were performed to evaluate the association between morphological values and MPL grade. Thus, MPL grade 1, 2, and 3 were included in the results.

S3 Table: Please check the accuracy. Two tables are the same (PPP and DPP).

Response: The VIF for PPP and the VIF for DPP were the same because the final model for PPP and DPP had the same independent variables. For example, the VIF for joint angle in the final model for PPP with anatomical values (the first section of S3 Table) was calculated using coefficients of determination for joint angle with age, PLL/PL, AT angle, TL/PL, and TL/FC. It doesn’t involve a dependent variable (in this case PPP); thus, the resulting tables for the final models for PPP and DPP became the same. However, I noticed that the table for PPP was inputted twice; hence, I deleted one of them. I apologize for the confusion.

Reviewer #2: Dear. the author

The authors evaluate Examination of the proximodistal patellar position in small dogs in relation to anatomical features of the distal femur and medial patellar luxation.

The reviewer suggests rejecting the article first of all because to gross methodological errors.

Kind regards,

Response:

Thank you for the comment. The merit of this study is to clarify the influence of anatomical factors of the distal femur on the proximodistal patellar position. We also clarified the association between MPL and proximodistal patellar position. Although PLL/PL is widely used to determine the proximodistal patellar position, our results showed that other morphological features also influence the proximodistal patellar position. Our study revealed that it might not be sufficient to determine the proximodistal patellar position by PLL/PL only. This paper also suggested that the proximodistal patellar position was not significantly different between dogs with MPL and dogs without MPL. This will give a new insight into the long-running dispute about the association between patella alta and MPL.

I could not identify the methodological errors being referred to as they were not specified, but I hope that our responses to the other reviewers have addressed your concerns.

Reviewer #3: The key findings in this study are

1) The proximodistal patellar position of small dogs depends on the size of the femoral condyle, the shape of the trochlea, and the length of patellar ligament.

2) A shorter patellar ligament, an upright trochlea, greater trochlear length, and bigger femoral condyle will lead to a distally positioned patella.

3) The proximodistal patellar position is not significantly different between the hindlimbs with and without MPL.

The main limitation with this study is that the authors included 20 dog breeds in their study and due to the fact that there are differences between the dog breeds regarding the measurements mentioned in this study, it will be difficult to get precise results.

Another limitation is that no consideration is given to the fact that flexion and extension of the joint can affect the position of the patella within the femoral trochlea. In this case, the joint angle measurement is irrelevant and should not be taken into account.

Response:

Thank you for your comments.

I added the breed specific data for toy poodles in lines 226-229. Although we could not analyse the intra-breed differences for every breed, considering the case groups as a mixed population, this study showed some significant results. I believe it is of equal importance to look at the differences in a mixed population as it is to look at the differences in a single breed of dogs. This was written in lines 356-360.

The joint flexion and extension were included in this study because I am fully aware it affects the position of the proximodistal patellar position. Thus, the stifle joint angle was included in the multiple regression as an independent variable. The coefficients and p-values for each model are shown in Tables 3, 4, 5, and 6. Joint angle was also included in the discussion, lines 308-316.

General Comment:

Where possible, avoid writing in the passive voice.

Rephrase sentences to be to the point. Avoid dragging out sentences by making it too wordy.

The overuse of prepositions/non-content words makes sentences hard to follow and unclear to understand. Consider rewriting/rephrasing where needed.

Response: Thank you for the comment. I shared this with my English language editor and rewrote some sentences as per your request.

34 Insert a comma before “and”

62 Consider replacing “On the basis of” with “Based on”

Response: I changed them as per your request.

66 Replace “and” with “or”

68 Replace “and” with “or”

Response: I compared dogs with MPL and dogs without MPL. Thus, “and” cannot be replaced with “or”.

70 – 77 Change the tense of the objective and hypothesis to present tense (change “was” to “is”)

80 Consider removing “that were”

85 The meaning of the sentence might be unclear. Consider using “or” instead of “and.” Alternatively, consider using a comma before “and”

Response: I changed them as per your request.

90-92 Consider rewriting the sentence, using fewer non-content words.

Response: I rewrote them as “Radiographs of hindlimbs with MPL were classified into the MPL group and radiographs of hindlimbs without any orthopaedic disease were classified into the control group.” Please check lines 95-97.

95 Consider rewriting the sentence starting with: “Based on a previous study…”

Response: I changed them as per your request.

111-112 The values of the joint angle cannot be constant in all measured x-rays as it depends on the flexion and extension position of the joint. It should not be taken into account in this study

Response: It was taken into account because the stifle joint angle was different among all x-rays. As the stifle joint angle was included in the multivariable models as a dependent variable, the coefficients of other variables can be interpreted as a rate of change when the stifle joint angle is constant.

112-113 To ensure accurate anatomical femoral axis, you need to take the whole femoral shaft into consideration. See Aiyan et al. “Measurement of the femoral neck angle in medium and large dog breeds” Acta Veterinaria Hungarica 67 (1), pp. 22–33 (2019). DOI: 10.1556/004.2019.003

Response: The anatomical axis used here is of the distal femur, not of the whole femur, which was used in the study you mentioned. The anatomical axis of the femur is not a straight line; if you want to draw it as a straight line, it would only be an estimated line. If the whole femoral shaft is to be considered, you can estimate the line of anatomical axis of the whole shaft. However, I used the anatomical axis of the distal femur, for which I decided to use the points I described in this manuscript to estimate it as a straight line.

See Fox DB, Tomlinson JL. Chapter 47 - Principles of angular limb deformity correction. Veterinary surgery: small animal. 2012;1:657-68.

113 Consider labelling the two points (white squares).

Response: I labelled them as A and B. Please check lines 125-126.

113 I think you mean the “distal point” not “proximal point”.

115 I think you mean the “proximal point” not “distal point”. 

116 “distal point”. The “proximal” tibial anatomical axis

Response: I corrected them accordingly. Thank you for the advice.

110-121 General comment regarding Fig.1: I suggest, providing a detailed description of the method used to identify the anatomical axis of the femur and tibia in the M&M and provide the references for all measurement methods. In this case, include only the labels in the figure legend

123 Fig.2 you already mention the definitions in the M&M (avoid repetition)

Response: I added a methodological description in the M&M. Please check lines 100-121. However, since I wanted the method to be understandable even for people who only read the figure legends, I decided to retain the figure legends.

130-135 The distal white line in fig. 3 is located few mm below/distal to the origin of the long digital extensor muscle! Provide references for the calculation of the PPP and DPP.

Response: I located the distal white line a few mm distal because the black line and the white line were almost at the same level. However, I changed the image for Fig 3. Kindly check.

All the references I used to measure the values were cited in the M&M. Please check lines 118-121.

142 As mentioned previously, the joint angle depends on the flexion and extension position of the joint and it cannot be constant in all studied dogs. (I think, you have to exclude it from your statistical analyses)

Response: The stifle joint angle was included in this study as it affects the proximodistal patellar position. The stifle joint angle, as well as all the other values like PLL/PL, was different among all x-rays. Because the stifle joint angle was included in the multivariable models as a dependent variable, the coefficients of other variables can be interpreted as a rate of change when the stifle joint angle is constant. We consulted with an expert for our statistical analyses.

152 This sentence might be unclear, consider rephrasing it. Ensure the correct use of “or” vs “and”

Response: The use of “or” and “and” is correct. 

175 Consider rewriting the sentence. Remain consistent: “1 Beagle, 1 Cavalier King…” or “a Beagle, a Cavalier King…”

Response: I rewrote them as per your request. Please check lines 194-199.

180 Insert a comma before “and”

Response: I did not insert a comma because there would be too many commas in one sentence. A comma is not required after “53 were left limbs” because the clause that follows is short and closely related.

191 Including the stifle joint angle will confuse the readers for the same reason mentioned previously. See the large SD range

Response: The SD range for the stifle joint angle was large because the angle value was large. In comparing dispersion of values, coefficient of variation (SD divided by mean) must be used. The coefficients of variation for stifle joint angle are 0.13 for the MPL group and 0.16 for the control group. They are not much larger than those of the PLL/PL, which are 0.12 for the MPL group and 0.11 for the control group. This cannot be the reason for excluding the stifle joint angle from the statistical analysis.

210 Consider rephrasing this sentence.

Response: I rephrased it as “For sex, spayed female, male, and female were compared with castrated males. Limb side was the right side compared with the left side.” Please check lines 236-237.

223 The phrase “control group” is missing a determiner: “a” or “the”

Response: I added “the” before “control group”. Please check line 250.

233 The sentence is missing a full stop at the end.

Response: I apologize, but I cannot seem to understand what you mean. Please clarify further so we I can address your concern appropriately.

239 Change “were” to “was”

Response: I used “were” because there are two subjects in this sentence. FC/PL and TL/PL. Please check lines 263-264.

250 The phrase “control group” is missing a determiner: “a” or “the”

Response: I added “the” before “control group”. Please check line 275.

256 “variance” is missing a determiner: “a”

Response: I rewrote it as per your request. Please check line 289.

279 Consider rephrasing to: “the long femoral axis”

Response: “The long femoral axis” is not found in articles in PubMed, whereas “the femoral long axis” is found in 7 articles. However, I rephrased it to “the long axis of femur” to avoid confusion. Please check lines 320-323.

280 Consider rephrasing to: “The TL/FC results…”

Response: I rephrased them to “The results of TL/FC on PPP and DPP indicate that…” Please check lines 323-325.

289 Insert a comma before and after the interruption: “…features, as well as PLL/PL, had…”

Response: I added it as per your request. Please check line 333.

307 Consider removing the word: “some”

Response: I removed it as per your request. Please check lines 350-351.

310 Remove the word: “a”

Response: I deleted it as per your request. Please check line 354.

314 Remove the word: “such”

Response: I edited it as per your request. Please check lines 357-359.

323 The phrase “patellar ligament” is missing a determiner: “the patellar ligament.”

Response: I added “the” before “patellar ligament”. Please check lines 365-367.

---

## [Decision Letter · Decision Letter 1]

18 May 2021

Examination of the proximodistal patellar position in small dogs in relation to anatomical features of the distal femur and medial patellar luxation

PONE-D-20-37854R1

Dear Dr. Murakami,

We’re pleased to inform you that your manuscript has been judged scientifically suitable for publication and will be formally accepted for publication once it meets all outstanding technical requirements.

Kind regards,

Silvia Sabattini

Academic Editor

PLOS ONE

Additional Editor Comments (optional):

Reviewers' comments:

Reviewer's Responses to Questions

**Comments to the Author**

1. If the authors have adequately addressed your comments raised in a previous round of review and you feel that this manuscript is now acceptable for publication, you may indicate that here to bypass the “Comments to the Author” section, enter your conflict of interest statement in the “Confidential to Editor” section, and submit your "Accept" recommendation.

Reviewer #1: All comments have been addressed

Reviewer #3: All comments have been addressed

2. Is the manuscript technically sound, and do the data support the conclusions?

Reviewer #1: Yes

Reviewer #3: Yes

3. Has the statistical analysis been performed appropriately and rigorously? 

Reviewer #1: I Don't Know

Reviewer #3: I Don't Know

4. Have the authors made all data underlying the findings in their manuscript fully available?

Reviewer #1: Yes

Reviewer #3: Yes

5. Is the manuscript presented in an intelligible fashion and written in standard English?

Reviewer #1: Yes

Reviewer #3: Yes

6. Review Comments to the Author

Reviewer #1: In the study named "Examination of the proximodistal patellar position in small dogs in relation to anatomical features of the distal femur and medial patellar luxation", the authors considered all the suggestions and explained them all. They made corrections where necessary in the text.

Reviewer #3: (No Response)

7. PLOS authors have the option to publish the peer review history of their article (what does this mean?). If published, this will include your full peer review and any attached files.

Reviewer #1: No

Reviewer #3: No

---

## [Editor Report · Acceptance letter]

20 May 2021

PONE-D-20-37854R1 

Examination of the proximodistal patellar position in small dogs in relation to anatomical features of the distal femur and medial patellar luxation 

Dear Dr. Murakami:

I'm pleased to inform you that your manuscript has been deemed suitable for publication in PLOS ONE. Congratulations! Your manuscript is now with our production department. 

Kind regards, 

on behalf of

Dr. Silvia Sabattini 

Academic Editor

PLOS ONE